# Three-Dimensional Microwave Imaging: Fast and Accurate Computations with Block Resolution Algorithms

**DOI:** 10.3390/s20216282

**Published:** 2020-11-04

**Authors:** Corentin Friedrich, Sébastien Bourguignon, Jérôme Idier, Yves Goussard

**Affiliations:** 1Laboratoire des Sciences du Numérique de Nantes, École Centrale de Nantes, 1 rue de la Noë, 44321 Nantes, France; Corentin.Friedrich@gmail.com (C.F.); Jerome.Idier@ls2n.fr (J.I.); 2École Polytechnique de Montréal, C.P. 6079, Succursale Centre-Ville, Montréal, QC H3C 3A7, Canada; yves.goussard@polymtl.ca

**Keywords:** microwave imaging, inverse scattering, numerical optimization, block system inversion

## Abstract

This paper considers the microwave imaging reconstruction problem, based on additive penalization and gradient-based optimization. Each evaluation of the cost function and of its gradient requires the resolution of as many high-dimensional linear systems as the number of incident fields, which represents a large amount of computations. Since all such systems involve the same matrix, we propose a block inversion strategy, based on the block-biconjugate gradient stabilized (BiCGStab) algorithm, with efficient implementations specific to the microwave imaging context. Numerical experiments performed on synthetic data and on real measurements show that savings in computing time can reach a factor of two compared to the standard, sequential, BiCGStab implementation. Improvements brought by the block approach are even more important for the most difficult reconstruction problems, that is, with high-frequency illuminations and/or highly contrasted objects. The proposed reconstruction strategy is shown to achieve satisfactory estimates for objects of the Fresnel database, even on the most contrasted ones.

## 1. Introduction

Microwave imaging aims at estimating the dielectric properties (permittivity and conductivity) of objects illuminated by incident electromagnetic fields [1]. This technique has been used in a wide range of applications such as biomedical imaging [2,3], geophysics [4,5] and nondestructive testing [6,7]. Quantitative inverse scattering methods estimate the complex permittivity map inside the object from a set of measured scattered electromagnetic fields around it. When the object size is comparable to the wavelength, diffraction occurs and the forward scattering model, which defines the scattered field as a function of the dielectric properties at any point of the object, becomes non-linear. The resulting reconstruction problem is ill-posed [8,9], and its resolution becomes particularly difficult with high-frequency illuminations and/or for highly contrasted objects—that is, if the permittivity inside the object strongly differs from that of the background—where the Born approximation is not valid anymore.

Imaging methods usually rely on solving an optimization problem, whose complexity is strongly impacted by the non-linearity of scattering. In order to circumvent this difficulty, several works introduced additional variables, corresponding to the total fields inside the object, which are jointly estimated together with the contrast function (i.e., the relative complex permittivity). Within this formulation, the domain equation, which links the total fields to the contrast function, is not exactly solved, but the resulting bilinear model allows the low-cost computation of iterates in the optimization procedure. The Modified Gradient Method [10] and Contrast Source Inversion methods [11,12] fall within this category, which achieve satisfactory compromise between computation time and reconstruction quality in cases where diffraction remains limited [13]. Although some works addressed 3-D reconstruction based on this formulation [7,14,15,16], most of them were limited to 2-D problems, where the number of unknowns remained relatively small [13,17,18,19,20,21,22,23,24].

A second class of methods, which dominates in the recent 3-D microwave imaging literature [25,26,27,28,29,30,31,32,33,34,35]. directly incorporates the domain equation into the problem formulation. The price to pay for using such an exact form of the domain equation is a much more complex evaluation of the cost function and of its gradient: both require a numerical solver for computing the forward problem, based on the resolution of very large linear systems, especially in the 3-D case. The related computational effort then represents the major cost of the reconstruction procedure. Therefore, developing efficient strategies for such system resolutions remains one of the current challenges for improving the efficiency of microwave imaging techniques [33,36,37]. In order to reduce the size of the problem, some works focused on the reconstruction of strongly constrained, parameterized, objects. Efficient numerical methods were developed with global convergence properties, e.g., based on evolutionary algorithms [38] or Markov Chain Monte-Carlo methods, coupled with surrogate models [39]. In the particular case of point-like scatterers, the two-step procedure in [40] efficiently accounts for the sparsity of the object.

In this paper, we focus on methods that aim to estimate any permittivity distribution (that is, without any strong prior assumption on the inspected object), based on the exact problem formulation. Previous works based on multiplicative [33] or additive [13], edge-preserving or sparsity-enhancing [41], regularization fall within this category. All these methods rely on repeated computations of forward scattering problems. One generally resorts to iterative system solving algorithms, which are stopped as soon as an acceptable residual error is obtained. Several such algorithms have been used in microwave imaging, e.g., the conjugate gradient (CG) algorithm [42,43], the biconjugate gradient (BiCG) algorithm [44], the quasi-minimal residual (QMR) algorithm [45,46] or the biconjugate gradient stabilized (BiCGStab) algorithm [33,36,37,47]. BiCGStab is now one of the most popular algorithms in this field and it was shown to converge faster than CG, BiCG and QMR [47].

In microwave imaging, as many forward scattering models as the number of illuminating sources must be computed. Such computations are usually performed sequentially or via massively parallel procedures [33,48]. Note that all such linear systems involve the same system matrix, which only depends on the object dielectric properties and on the discretization scheme. A classical solution relies on the preliminary LU-decomposition of the system matrix [4]. However, this possibility becomes prohibitive in 3-D imaging [49]. Block iterative algorithms specifically address the resolution of such linear systems with multiple right-hand sides (RHSs), by jointly solving all systems. Block versions of several standard iterative methods have been proposed in the applied mathematics literature, including the block-conjugate (and bi-conjugate) gradient algorithms [50], the block-QMR algorithm [51] and the block-BiCGStab algorithm [52]. Such block algorithms were shown to achieve better performance than their sequential counterparts in generic contexts. Block-QMR has been used in microwave imaging [49]. The block-BiCGStab algorithm proposed in [52] was shown to outperform block-QMR. Its standard implementation, however, is rather unsatisfactory for microwave imaging forward problems, where the system matrices are much larger, and where there are many more RHS terms which are highly correlated [53].

Here, we propose a reconstruction procedure based on additive penalization, where the evaluation of both the cost function and its gradient are performed with a dedicated block-BiCGStab implementation. Specific tunings of the block-BiCGStab algorithm are discussed, and the overall efficiency of the inversion method is evaluated as a function of the problem difficulty (illuminating frequency and contrast). We show in particular that improvements are even greater as the problem complexity increases. We provide reconstruction results obtained on the Fresnel database [54], where the proposed implementation allows the accurate reconstruction of the most difficult objects.

In Section 2, the 3-D forward scattering model is built using an integral formulation. A discretization procedure based on the method of moments is used, and our inversion formulation based on additive regularization is detailed. In Section 3, we propose efficient adaptations of block-BiCGStab for forward scattering computations, and for their implementation within the reconstruction procedure. In Section 4, algorithmic performance is evaluated with simulated data, as a function of the object complexity, with different illuminating frequencies and contrast levels. In Section 5, our method is applied to real measurements extracted from the Fresnel database [54], where reconstruction quality and computation times are discussed. Concluding comments are given in Section 6.

## 2. Formulation of 3-D Forward and Inverse Scattering Problems

### 2.1. Forward Scattering Problem

The forward scattering model defines the total electric field in a given volume *V* containing the scattering object, as a function of the spatial distribution of the object permittivity. We adopt a standard approach based on the integral formulation of the domain equation [1], followed by a discretization procedure based on the method of moments [55]—see for example [14,36,37,43,44] for similar choices. We consider a background medium with constant complex permittivity $\epsilon_{b}$, containing the object under study with complex permittivity ϵ(r) where r denotes the vector of spatial coordinates. The object is illuminated by an incident wave with angular frequency ω. In the sequel, the time dependency exp(−jωt) of all quantities is omitted. The electric field integral equation gives an implicit relation between the unknown total electric field at any point inside the domain under study and the object permittivity distribution [1]:(1)$${\vec{E}}^{\mathrm{tot}}\left(r\right)={\vec{E}}^{\mathrm{inc}}\left(r\right)+\left(k_{b}^{2}+\nabla \nabla \cdot \right)\int_{r^{\prime }\in V}g\left(r-r^{\prime }\right)\chi \left(r^{\prime }\right){\vec{E}}^{\mathrm{tot}}\left(r^{\prime }\right)dr^{\prime },\;\;\forall r\in V,$$
where ${\vec{E}}^{\mathrm{tot}}\left(r\right)$ and ${\vec{E}}^{\mathrm{inc}}\left(r\right)$ respectively denote the total and the known incident 3-D electric field vectors at point r∈*V*, $\mu_{0}$ is the vacuum permeability, $k_{b}=\sqrt{\omega^{2}\mu_{0}\epsilon_{b}}$ is the wavenumber in the background medium, $g\left(r\right)=\exp(jk_{b}∥r∥)/4\pi ∥r∥$ is the Green function and $\chi \left(r\right)=\left(\epsilon \left(r\right)-\epsilon_{b}\right)/\epsilon_{b}$ is the contrast function. The term ∇∇· represents the gradient of the divergence with respect to r.

Using the discretization procedure based on the method of moments (see [14] for the 3-D case), the volume *V* is divided into *N* cubic voxels and the integral Equation (Equation 1) turns into:(2)$$e^{\mathrm{tot}}=e^{\mathrm{inc}}+G_{D}Xe^{\mathrm{tot}}.$$

The size-3N vectors $e^{\mathrm{inc}}$ and $e^{\mathrm{tot}}$ respectively contain the three spatial components of the 3-D incident and total electric field vectors, discretized on the *N* voxels that are sorted according to the lexicographic order. The size-*N* vector x corresponds to the contrast function discretized on the *N* voxels, put in vector form, and X is the 3N×3N diagonal matrix whose diagonal contains three replicas of x. The 3N×3N matrix $G_{D}$ corresponds to the discretization of the 3×3 Green tensor $\left(k_{b}^{2}I_{3}+\nabla \nabla \right)g\left(r-r^{\prime }\right)$ on the *N* voxels, where $I_{3}$ denotes the 3×3 identity tensor (see for example [53] for details). From (Equation 2), the discretized total field is the solution to the following linear system:(3)$$L_{x}e^{\mathrm{tot}}\;=\;e^{\mathrm{inc}},\;\;\mathrm{with}\;\;L_{x}\;=\;I_{3N}\;-G_{D}X,$$
where $I_{3N}$ denotes the 3N×3N identity matrix.

In microwave imaging, the object is successively illuminated with a set of $N_{S}$ sources at different spatial locations, creating incident fields $e_{i}^{\mathrm{inc}},i=1,\ldots ,N_{S}$. Then, computing the corresponding total fields $e_{i}^{\mathrm{tot}}$ amounts to solving $N_{S}$ linear systems $L_{x}e_{i}^{\mathrm{tot}}\;=\;e_{i}^{\mathrm{inc}}$, that is:(4)$$L_{x}E^{\mathrm{tot}}=E^{\mathrm{inc}},\;\;\mathrm{with}\;\;E^{\mathrm{tot}}=\left[e_{1}^{\mathrm{tot}},\ldots ,e_{N_{S}}^{\mathrm{tot}}\right]\;\;\mathrm{and}\;\;E^{\mathrm{inc}}=\left[e_{1}^{\mathrm{inc}},\ldots ,e_{N_{S}}^{\mathrm{inc}}\right],$$
where all linear systems involve the same system matrix $L_{x}$. Such a property is the key point for our block resolution strategy.

### 2.2. Observation Model

The scattered field ${\vec{E}}^{\mathrm{scat}}$ at any point r outside *V* is defined by the volume integral observation equation:(5)$${\vec{E}}^{\mathrm{scat}}\left(r\right)=\left(k_{b}^{2}+\nabla \nabla \cdot \right)\int_{r^{\prime }\in V}g\left(r-r^{\prime }\right)\chi \left(r^{\prime }\right){\vec{E}}^{\mathrm{tot}}\left(r^{\prime }\right)dr^{\prime }.$$
Using the same discretization procedure as in Section 2.1, considering a set of $N_{M}$ measurement points, it becomes:(6)$$e^{\mathrm{scat}}=G_{O}Xe^{\mathrm{tot}},$$
where the size-$N_{M}$ vector $e^{\mathrm{scat}}$ collects the measurements of the scattered fields at the receiver locations, and the $N_{M}\times 3N$ matrix $G_{O}$ corresponds to the discretization of the Green tensor $\left(k_{b}^{2}I_{3}+\nabla \nabla \right)g\left(r-r^{\prime }\right)$, for $r^{\prime }$ at the *N* voxels in *V* and r at the measurement points [56]. Combining both domain (Equation 3) and observation (Equation 6) equations finally yields the observation model:(7)$$e_{i}^{\mathrm{scat}}\;=\;G_{O}XL_{x}^{-1}e_{i}^{\mathrm{inc}}=G_{O}X{\left(I_{3N}-G_{D}X\right)}^{-1}e_{i}^{\mathrm{inc}},\;\mathrm{for}\;\mathrm{any}\;\mathrm{incident}\;\mathrm{field}\;e_{i}^{\mathrm{inc}}.$$

### 2.3. Inverse Problem Formulation

The microwave imaging inverse problem consists of estimating the contrast function from the measured scattered fields at the receivers. This problem is ill-posed [8,9], notably because the relation between the contrast function and the data, obtained from the two coupled Equations (Equation 1) and (Equation 5) in the continuous formulation (Equations (Equation 2) and (Equation 6) in the discrete case), is highly nonlinear.

Here, we adopt a classical approach to inverse problems [13,57], which consists of minimizing the misfit between the data ${\tilde{e}}_{i}^{\mathrm{scat}}$ and the simulated scattered fields $e_{i}^{\mathrm{scat}}$, penalized by an additive regularization term. More precisely, we define the reconstructed contrast function $\hat{x}$ by:(8)$$\hat{x}=\underset{x}{\arg\;\min}F\left(x\right),\;\mathrm{with}\;F\left(x\right)=\frac{1}{2}\sum_{i=1}^{N_{S}}{\left|\left|{\tilde{e}}_{i}^{\mathrm{scat}}-G_{O}XL_{x}^{-1}e_{i}^{\mathrm{inc}}\right|\right|}^{2}+R\left(x\right),$$
where the regularization term operates on differences of the contrast function at neighboring voxels:(9)$$R\left(x\right)=\gamma \sum_{i,j,\;i∼j}\varphi \left(x_{i}-x_{j}\right)\;\mathrm{with}\;\gamma >0,$$
where notation i∼j selects indices of neighbor voxels and φ is the “$\ell_{2}\ell_{1}$” function $\varphi \left(u\right)=\sqrt{\delta^{2}+u^{2}}$, where δ is a small positive constant ensuring the differentiability of the cost function, which was set to $10^{-2}$ in all our experiments. Such a penalization favors edge-preserving solutions and leads to a smooth cost function F. Therefore, optimization can be addressed with gradient-based iterative algorithms. The penalization weight γ trades off between fidelity to data and regularization, which is necessary since the problem is ill-posed [8,9]. Its value may depend on the noise level and on the problem size.

Note that alternate regularization strategies, such as multiplicative regularization [33,58], have also been proposed. Here, we chose additive regularization because it was previously shown to be particularly efficient in the case of highly contrasted objects [13]. However, the techniques presented in this paper may also be applied to reconstruction methods based on multiplicative regularization, since both the corresponding objective function and its gradient present similar algebraic characteristics.

### 2.4. Optimization and Computational Issues

We propose to solve the optimization problem (Equation 9) with the L-BFGS algorithm, which is an acknowledged method for solving high-dimensional convex optimization problems  ([59], see for example [13] in application to microwave imaging). Each iteration of L-BFGS consists of two steps: computing a descent direction, based on the computation of the gradient at the current point, and finding a convenient step size, for which several evaluations of the cost function may be necessary. The gradient of F reads:(10)$$\nabla F\left(x\right)=-Q\sum_{i=1}^{N_{S}}J_{i}^{†}\left({\tilde{e}}_{i}^{\mathrm{scat}}-G_{O}XL_{x}^{-1}e_{i}^{\mathrm{inc}}\right)+\nabla R\left(x\right)$$
where $Q=[I_{N},I_{N},I_{N}]$, the symbol ${}^{†}$ denotes the conjugate transpose and $J_{i}$ is the Jacobian matrix associated to the *i*-th data misfit term in (Equation 8). It can easily be shown that:(11)$$J_{i}=G_{O}{\left(L_{x}^{T}\right)}^{-1}\mathrm{diag}\left\{L_{x}^{-1}e_{i}^{\mathrm{inc}}\right\},$$
where diag{u} stands for the diagonal matrix with diagonal u, such that the gradient reads:(12)$$\nabla F\left(x\right)=-Q\sum_{i=1}^{N_{S}}\mathrm{diag}{\left\{L_{x}^{-1}e_{i}^{\mathrm{inc}}\right\}}^{†}{\bar{L_{x}}}^{\;-1}G_{O}^{†}\left({\tilde{e}}_{i}^{\mathrm{scat}}-G_{O}XL_{x}^{-1}e_{i}^{\mathrm{inc}}\right)+\nabla R\left(x\right),$$
where $\bar{L_{x}}$ denotes the conjugate of $L_{x}$.

Inspection of (Equation 8) and (Equation 12) reveals that the bulk of the computational effort required for evaluating F and ∇F lies in the evaluation of matrix-vector products involving matrices X and $G_{O}$, and in the resolution of linear systems with matrices $L_{x}$ and $\bar{L_{x}}$. Note that $G_{O}$ is an $N_{M}\times 3N$ matrix and this manageable size makes it possible to explicitly store it for direct computations; similarly, X is a 3N×3N diagonal matrix, which results in small memory and computational requirements. Indeed, the major difficulty lies in the resolution of the $N_{S}$ systems of size 3N×3N involving matrix $L_{x}$ (computation of $L_{x}^{-1}e_{i}^{\mathrm{inc}}$), and in the resolution of another $N_{S}$ similar systems involving matrix $\bar{L_{x}}$ (computation of ${\bar{L_{x}}}^{\;-1}v_{i}$, with $v_{i}=G_{O}^{†}\left({\tilde{e}}_{i}^{\mathrm{scat}}-G_{O}XL_{x}^{-1}e_{i}^{\mathrm{inc}}\right)$). Note that, in the frequent case where the number of sources exceeds the number of receivers, by virtue of the Lorentz reciprocity theorem (see for example [60]), one can equivalently switch the roles of sources and receivers, then reducing the number of systems to be solved.

In Section 3, we propose a dedicated block resolution strategy in order to jointly solve such multiple systems. This iterative method requires the computation of high-dimensional matrix-vector products involving matrix $L_{x}$ (resp. $\bar{L_{x}}$), that is, involving the 3N×3N matrix $G_{D}$ (resp. $\bar{G_{D}}$). Note that these products are convolution products, which can be efficiently computed with 3-D Fast Fourier Transform (FFT) algorithms [36,47].

## 3. Simultaneous Resolution of Multiple Forward Scattering Problems

We now address the joint resolution of the multiple forward scattering problems (Equation 4) by the block-BiCGStab algorithm proposed in [52]. We present the algorithm and we focus on implementation issues for solving one set of multiple scattering problems.

### 3.1. Description of the Block-BiCGStab Algorithm

The pseudo-code implementing the block-BiCGStab algorithm for the joint resolution of (Equation 4) is given in Algorithm 1. The notation ${\langle T,S\rangle }_{F}$ represents the Frobenius product of two matrices. For comparison purposes, the sequential implementation with the BiCGStab algorithm, commonly used in microwave imaging, is recalled in Algorithm 2.
**Algorithm 1** Joint resolution of $N_{S}$ linear systems $L_{x}E^{\mathrm{tot}}=E^{\mathrm{inc}}$ with block-BiCGStab.1:For an initial matrix guess ${E^{\mathrm{tot}}}^{\left(0\right)}$,set $R^{\left(0\right)}=E^{\mathrm{inc}}-L_{x}{E^{\mathrm{tot}}}^{\left(0\right)}$, $P^{\left(0\right)}=R^{\left(0\right)}$  2:Choose an arbitrary $3N\times N_{S}$ matrix ${\tilde{R}}_{0}$  3:k=0  4:**repeat**5: $V=L_{x}P^{\left(k\right)}$  6: solve $\left({\tilde{R}}_{0}^{†}V\right)A={\tilde{R}}_{0}^{†}R^{\left(k\right)}$  7: $S=R^{\left(k\right)}-VA$  8: $T=L_{x}S$  9: $\omega ={\langle T,S\rangle }_{F}/{\langle T,T\rangle }_{F}$  10: ${E^{\mathrm{tot}}}^{\left(k+1\right)}={E^{\mathrm{tot}}}^{\left(k\right)}+P^{\left(k\right)}A+\omega S$  11: $R^{\left(k+1\right)}=S-\omega T$  12: solve $\left({\tilde{R}}_{0}^{†}V\right)B=-{\tilde{R}}_{0}^{†}T$  13: $P^{\left(k+1\right)}=R^{\left(k+1\right)}+\left(P^{\left(k\right)}-\omega V\right)B$  14: k←k+1  15:**until**$\forall i,∥r_{i}^{\left(k\right)}∥/∥e_{i}^{\mathrm{inc}}∥<\mathrm{tolerance}$

**Algorithm 2** Sequential resolution of $N_{S}$ linear systems $L_{x}e_{i}^{\mathrm{tot}}=e_{i}^{\mathrm{inc}}$, $i=1,\ldots ,N_{S}$, with BiCGStab.
0:
**for**
$i=1,\ldots ,N_{S}$
**do**
1: For an initial guess ${e_{i}^{\mathrm{tot}}}^{\left(0\right)}$,  set $r^{\left(0\right)}=e_{i}^{\mathrm{inc}}-L_{x}{e_{i}^{\mathrm{tot}}}^{\left(0\right)}$, $p^{\left(0\right)}=r^{\left(0\right)}$  2: Choose an arbitrary vector ${\tilde{r}}_{0}$  3: k=0  4: **repeat**5:  $v=L_{x}p^{\left(k\right)}$  6:  $\alpha =\left({\tilde{r}}_{0}^{†}r^{\left(k\right)}\right)/\left({\tilde{r}}_{0}^{†}v\right)$  7:  $s=r^{\left(k\right)}-\alpha v$  8:  $t=L_{x}s$  9:  $\omega =t^{†}s/t^{†}t$  10:  ${e_{i}^{\mathrm{tot}}}^{\left(k+1\right)}={e_{i}^{\mathrm{tot}}}^{\left(k\right)}+\alpha p^{\left(k\right)}+\omega s$  11:  $r^{\left(k+1\right)}=s-\omega t$  12:  $\beta =\left(\alpha {\tilde{r}}_{0}^{†}r^{\left(k+1\right)}\right)/\left(\omega {\tilde{r}}_{0}^{†}r^{\left(k\right)}\right)$  13:  $p^{\left(k+1\right)}=r^{\left(k+1\right)}+\beta \left(p^{\left(k\right)}-\omega v\right)$  14:  k←k+1  15: **until**
$∥r^{\left(k\right)}∥/∥e_{i}^{\mathrm{inc}}∥<\mathrm{tolerance},$  16:
**end for**



In both algorithms, the most time-consuming operations concern the computation of matrix-vector products of the form $L_{x}p=p-G_{D}Xp$ in steps 5 and 8. Note that the same number ($2N_{S}$) of such products are performed by one iteration of block-BiCGStab and $N_{S}$ iterations of BiCGStab. In the remaining steps, a set of $N_{S}$ dot products and scalar-vector products (one per system inversion) in BiCGStab are replaced by one $N_{S}$-dimensional matrix-vector product in block-BiCGStab. Similarly, a set of $N_{S}$ scalar divisions are replaced by one $N_{S}\times N_{S}$ system inversion (steps 6 and 12). Therefore, such block operations generate a slight extra cost in block-BiCGStab compared to BiCGStab. Because of the small size of the involved systems, this remains negligible compared to the computation time required by multiplications with $L_{x}$. Most of all, block operations introduce some coupling in the auxiliary variables and, subsequently, in the directions that are computed in order to update the solution at step 10. The rationale behind block versions is that coupling the descent directions yields more efficient update steps, and therefore results in a decrease in the total number of iterations and computing time.

### 3.2. Efficient Implementation for Solving Multiple Forward Problems

In its original description [52], the block-BiCGStab algorithm was evaluated on problems involving only a small number (5 to 10) of RHS vectors which were drawn independently, in moderate dimensions (up to 2500×2500). In microwave imaging, the number of RHS vectors—the number of measurements—is much higher, and vectors are highly correlated, since they correspond to the different incident illuminating fields, which are spatially close to each other. Moreover, matrices involved in 3-D problems are usually much bigger than those used in [52].

Numerical experiments, whose salient results are reported in Section 4.2, revealed that the conditioning of matrix ${\tilde{R}}_{0}^{†}V$, which enters steps 6 and 12 of Algorithm 1, has a critical impact on the behavior and convergence of the block-BiCGStab algorithm. Inspection of steps 1 and 5 clearly indicates that the conditioning of V strongly depends on that of $R^{\left(0\right)}$. With the standard initialization ${E^{\mathrm{tot}}}^{\left(0\right)}=E^{\mathrm{inc}}$, when two sources *i* and *j* are close, the corresponding incident fields $e_{i}^{\mathrm{inc}}$ and $e_{j}^{\mathrm{inc}}$ which make up the corresponding columns of ${E^{\mathrm{tot}}}^{\left(0\right)}$, become highly correlated; hence, $R^{\left(0\right)}$ becomes poorly conditioned and ${\tilde{R}}_{0}^{†}V$ is near-singular, which may lead to numerical errors propagating in the computations of matrices A and B (steps 6 and 12). This pathological behavior does not occur in the simulated experiments reported in [52], since only statistically independent RHS vectors are considered.

Here, we propose to add a small random quantity to ${E^{\mathrm{tot}}}^{\left(0\right)}$ in the initialization step 1. This reduces the correlation between columns of ${E^{\mathrm{tot}}}^{\left(0\right)}$, and thus improves the conditioning of matrix ${\tilde{R}}_{0}^{†}V$. Empirically, we have found that an initial perturbation with a signal-to-noise ratio ranging from 40 to 100 dB yielded roughly the same favorable effect, while a higher level of perturbation would slow down the convergence, and a smaller one would amount to no perturbation. In all experiments in this paper including such noisy initialization, the signal-to-noise ratio was set to 50 dB. The results reported in Section 4.2 show that the algorithm behaves in a satisfactory manner with such an initialization. Regarding the choice of matrix ${\tilde{R}}_{0}$, various options were studied; the best results were obtained by using the choice proposed in [52], that is, ${\tilde{R}}_{0}=R^{\left(0\right)}=E^{\mathrm{inc}}-L_{x}{E^{\mathrm{tot}}}^{\left(0\right)}$.

### 3.3. Implementation within the Reconstruction Procedure

We now consider the use of the block-BiCGStab algorithm for solving the two multiple linear systems involved in the evaluation of the cost function (Equation 8) and of its gradient (Equation 12) at each iteration of the optimization procedure. The first system reads $S^{\left(t\right)}:{E^{\mathrm{tot}}}^{\left(t\right)}=L_{x^{\left(t\right)}}^{-1}E^{\mathrm{inc}}$, and the second one is of the form ${\bar{S}}^{\left(t\right)}:V^{\left(t\right)}={\bar{L}}_{x^{\left(t\right)}}^{-1}U^{\left(t\right)}$, where notation ${}^{\left(t\right)}$ indexes the iteration number. According to the expression of the gradient (Equation 12), the RHS term $U^{\left(t\right)}$ in ${\bar{S}}^{\left(t\right)}$ depends on the solution of $S^{\left(t\right)}$, so the two systems can only be solved sequentially and not jointly. In the following, details are given considering $S^{\left(t\right)}$, but similar arguments hold for ${\bar{S}}^{\left(t\right)}$.

As discussed in Section 3.2, initialization of block-BiCGStab plays an important role in the algorithm efficiency. Our implementation is based on the fact that two consecutive iterates of the optimization algorithm are close to each other, especially when the reconstruction algorithm is near to convergence. Therefore, the solution ${E^{\mathrm{tot}}}^{\left(t+1\right)}$ should be close to that obtained at the former iteration ${E^{\mathrm{tot}}}^{\left(t\right)}$, and ${E^{\mathrm{tot}}}^{\left(t\right)}$ could be used as an initialization of block-BiCGStab for solving $S^{\left(t+1\right)}$. In particular, ${E^{\mathrm{tot}}}^{\left(t\right)}$ should be much closer to ${E^{\mathrm{tot}}}^{\left(t+1\right)}$ than the matrix formed by the incident fields $E^{\mathrm{inc}}$, that was used for initializing block-BiCGStab for the resolution of a single set of forward problems in Section 3.2. We noticed, however, that implementing this strategy led to poor convergence rates, because the total fields ${E^{\mathrm{tot}}}^{\left(t\right)}$ are highly correlated with each other, which generated conditioning issues similar to that explained in Section 3.2. Therefore, we propose to initialize iteration t+1 using a random perturbation added to ${E^{\mathrm{tot}}}^{\left(t\right)}$.

## 4. Performance of Block-BiCGStab on Synthetic Data

The results presented in this section were obtained on synthetic data representing a typical experimental setup. First, we illustrate the impact of the implementation choices discussed in Section 3.2. Then, we investigate the performance of the proposed block-BiCGStab implementation for computing multiple forward scattering problems as a function of the problem difficulty, by varying the object contrast and the illuminating frequency. Finally, the computational efficiency of block-BiCGStab is evaluated on the full reconstruction process.

### 4.1. Description of the Numerical Example and Implementation Details

The object used in our experiments was composed of two nested cubes embedded in air. This example was taken from [33] and the contrast was increased by a factor of two in order to make the reconstruction problem more challenging. The object domain *V* was a cube with a 30-cm edge size, which was discretized onto a 30×30×30 regular Cartesian grid ($1\;{\mathrm{cm}}^{3}$ voxels). It is usually acknowledged that the discretization grid must have at least four voxel sides per wavelength for accurate estimation of the scattered field. Here, the highest illuminating frequency is 3 GHz (see Section 4.3.1), that is, λ=10 cm, so that the voxel side is λ/10. The external part of the object was a cube with a 20-cm edge size, the contrast of which was set to $\chi_{1}=0.6+j0.8$. The internal part was a smaller cube with a 10-cm edge size and contrast $\chi_{2}=1.2+j0.4$. The object was illuminated by $N_{S}=160$ sources (vertical electric dipoles) that were regularly spaced around the object on five circles with a 60 cm diameter, at heights z=−20 cm, −10 cm, 0, 10 cm and 20 cm. Measurements were simulated by considering 160 receivers at the same locations as the sources. The object and the acquisition setup are represented in Figure 1.

Data were generated according to model (Equation 7) using a grid twice as fine as the one used in our numerical experiments (60×60×60 voxels) for more accurate computations, and circular Gaussian white noise was added in order to obtain a 20-dB SNR. Only the *z* component of the scattered field was used in our tests. Then, the reconstruction problem had $30^{3}$ = 27,000 complex unknowns and $160^{2}$ = 25,600 complex-valued data points.

All computations were performed with an Intel Core i7-5960X (eight cores) clocked at 3 GHz using Matlab, with multithreaded operations. For BiCGStab, a marching-on-in-source procedure [37] was used when it improved convergence. This procedure allows for better initialization of the solutions by using an interpolation based on the total fields previously computed for the nearest sources. The tolerance on the relative residuals imposed for convergence of both BiCGStab and Block-BiCGStab (line 15 in Algorithms 1 and 2) was set to $10^{-6}$. Therefore, the solutions obtained by the two algorithms are the same up to such numerical precision.

### 4.2. Impact of Initialization

The solution to the 3D forward problem corresponding to the experimental setup presented in Section 4.1 was computed with both BiCGStab and block-BiCGStab algorithms. First, the solutions were initialized to the incident fields (${e_{i}^{\mathrm{tot}}}^{\left(0\right)}=e_{i}^{\mathrm{inc}}$). Evolution of the iterated relative residuals $∥e_{i}^{\mathrm{inc}}-L_{x}{e_{i}^{\mathrm{tot}}}^{\left(k\right)}∥/∥e_{i}^{\mathrm{inc}}∥$ is shown in Figure 2a,b. BiCGStab converges in between 18 and 21 iterations for all problems, whereas block-BiCGStab residuals first slowly decrease, and then increase, leading to divergence.

Then, as described in Section 3.2, noise with 50 dB SNR was added to the initial solution of block-BiCGStab. The corresponding iterates are shown in Figure 2c—BiCGStab iterates, which are not represented, behave similarly to those represented in Figure 2a. On the contrary, block-BiCGStab iterates now converge much faster: in this example, block-BiCGStab converges in 40% less iterations than BiCGStab, corresponding to a similar saving in terms of computation time.

In all remaining experiments in the paper, the block-BiCGStab algorithm was initialized by adding such a noise to the incident fields.

### 4.3. Performance for Different Forward Scattering Configurations

We now compare BiCGStab and block-BiCGStab for several configurations of the 3-D forward problem based on the previous setup. The tests consisted of solving the $N_{S}$ systems $\left(I_{3N}-G_{D}X\right)e_{i}^{\mathrm{tot}}=e_{i}^{\mathrm{inc}}$, and we focused on the influence of the illuminating frequency and of the contrast values in X, since both factors impact the problem difficulty—the Born approximation, for example, is only valid for low-contrast objects with small electric size.

#### 4.3.1. Impact of the Frequency

We first study the algorithmic performance as a function of the illuminating frequency. The object defined in Section 4.1 was used, with frequencies equal to 1, 2 and 3 GHz. The corresponding wavelengths are respectively λ=30 cm, λ=15 cm and λ=10 cm, and the size of the reconstructed volume respectively corresponds to λ, 2λ and 3λ: the higher the frequency, the bigger the electric size of the object.

Table 1 gives the corresponding central processing unit (CPU) time and the number of iterations required for BiCGStab and block-BiCGStab to converge. First, in accordance with the discussion in Section 3.1, the CPU time is almost proportional to the number of iterations performed by both algorithms. Second, block-BiCGStab always yields a shorter computation time than BiCGStab, and the improvement becomes more significant as the frequency increases: at 3 GHz, block-BiCGStab is 33% faster than BiCGStab for solving the 160 systems.

#### 4.3.2. Impact of the Contrast

We now evaluate the computation time as a function of the contrast. A scaling factor ranging from 0.25 to 2.5 was applied to the synthetic object defined in Section 4.1. The illuminating frequency was set to 3GHz. Table 2 gives the corresponding CPU time and the number of iterations required for BiCGStab and block-BiCGStab. Here again, the CPU time is almost proportional to the number of iterations. In all cases, block-BiCGStab is faster than BiCGStab and, more importantly, the relative saving in CPU time increases with the contrast of the object: for a contrast factor of 2.5, block-BiCGStab runs approximately twice as fast as BiCGStab.

The results reported in Section 4.3.1 and Section 4.3.2 therefore suggest that block-BiCGStab is best suited for the resolution of difficult problems, i.e., involving large and/or highly contrasted objects.

### 4.4. Inversion Procedure and Object Reconstruction

We now consider the reconstruction problem with the simulation setup described in Section 4.1. Since the object was highly contrasted, illuminations at frequencies 1, 2 and 3 GHz were used and the following frequency hopping technique was implemented (see [28,61,62] for example):the initial solution was set to zero, and $N_{\mathrm{it}}$ iterations of the inversion algorithm were performed with the 1 GHz data;another $N_{\mathrm{it}}$ iterations were then performed with the 2 GHz data, with an initial solution set to the output of the previous step;the reconstruction algorithm was finally applied to 3 GHz data, with an initial solution set to the output of the previous step, and iterations are run until the $\ell_{\infty }$-norm of the gradient becomes lower than $10^{-6}$.

The number $N_{\mathrm{it}}$ of iterations for each intermediate frequency is a trade-off between reconstruction quality and computation time. In this experiment, we found that $N_{\mathrm{it}}=20$ iterations were sufficient to achieve satisfactory results (that is, more iterations at intermediate frequencies would not improve the final reconstruction and would unnecessarily increase the computation time). The regularization weight in (Equation 9) was set to $\gamma =10^{-6}$ for the three frequencies.

Reconstruction was performed using BiCGStab and block-BiCGStab, following the procedure proposed in Section 3.3. Since both algorithms used the same convergence thresholds, evaluations of the objective function and of its gradient at any point by the two algorithms were nearly identical; therefore, the minimization process followed roughly the same sequence, and the reconstructed objects obtained with both approaches are comparable. Reconstruction results are shown in Figure 3, together with the true object. The shapes of the two cubes are well retrieved and the contrast of the outer cube ($\chi_{1}=0.6+j0.8$) is very close to that of the true object. The boundaries of the reconstructed cubes slightly exceed the true ones, especially for the inner cube. The contrast value in the inner cube is also slightly misestimated (underestimation of the real part and overestimation of the imaginary part). Note that the inner cube is highly contrasted ($\chi_{2}=1.2+j0.4$), which makes it very difficult to reconstruct.

For this problem, the reconstruction lasted about 21.3 h when computations were performed with BiCGStab. The reconstruction time dropped to 16.8 hours when block-BiCGStab was used, which represents a saving of more than 20%.

## 5. Reconstruction of the Fresnel Database Objects

We finally consider the reconstruction of the five objects of the Fresnel database [54]. Such objects have already been reconstructed in many papers [27,28,29,61,63,64]—see also a summary of reconstruction results in [65]. Due to space limitations, we restrict the presentation to two objects, which can be considered as the two most difficult ones: the TwoSpheres and the Cylinder objects.

The 3-D experimental setup for the Fresnel data set is detailed in [54]. Using the reciprocity theorem and the two polarizations of the experimental data [29], we consider that the setup was equivalently composed of 36 sources placed on a circle at z=0 with a 1.796-m radius. Data were equivalently acquired with 81 receivers placed on a sphere with a 1.796-m radius [54]. The incident fields were plane waves polarized along the *z*-axis and acquisitions were performed at frequencies ranging from 3 to 8 GHz. The two polarizations were used (co- and cross-polarization). Since the measurements were performed in the far-field zone of the scattering object, the electric field at the receivers is transverse (that is, its longitudinal component is zero). When projected into the Cartesian coordinate system, the three spatial components of the measured field are generally non-zero. Thus, each data set is made up of 36×81×3=8748 complex-valued measurement points. For all objects, the volume of interest *V* was defined as a cube with a 10 cm edge size divided into 30×30×30 voxels, so that the object is fully enclosed in *V* in each case. The maximum frequency is 8 GHz, that is, λ=37.4 mm, so that the coarser discretization scheme corresponds to 11.3 voxel sides per wavelength. We then have 27,000 complex-valued unknowns. In all experiments, the regularization parameter γ in (Equation 9) was set to $\gamma =10^{-6}$.

Here also, as discussed in Section 4.4, a frequency hopping procedure was used to carry out the reconstructions as follows:Perform $N_{\mathrm{it}}$ iterations of the reconstruction algorithm with the 3 GHz data and initial contrast function set to zero.Perform $N_{\mathrm{it}}$ iterations of the reconstruction algorithm with the 4 GHz data and initial contrast function set to the final iterate of the previous step.Repeat the previous step with the 5 GHz, 6 GHz and 7 GHz data.Perform reconstruction of the object with the 8 GHz data, initial contrast function set to the final iterate of the previous step and a stopping rule on the $\ell_{\infty }$-norm of the gradient set to $10^{-6}$.

For all objects except Cylinder, computing $N_{\mathrm{it}}=10$ iterations at each frequency empirically led to a satisfactory compromise between reconstruction quality and time. For Cylinder, which was bigger and more contrasted, $N_{\mathrm{it}}=20$ intermediate iterations were necessary in order to achieve the best reconstruction.

The TwoSpheres object was composed of two identical spheres in contact embedded in air, 50 mm in diameter and with uniform, real-valued, relative permittivity equal to 2.6, that is, contrast χ=1.6. Figure 4 (top) shows 30 slices of the true object and of our corresponding reconstruction, and Figure 4 (bottom) represents the isosurface of the reconstruction at the contrast value χ=1. Note that we only present the real part of the estimated contrast function because the estimated imaginary part was very small. The latter is presented in the supplementary data files. It can be observed that the shape of the object was well reconstructed, even at the contact point. The contrast value was slightly overestimated at the contact point (the estimated contrast was about 1.8), and slightly underestimated around it (the estimated contrast was about 1.3).

The Cylinder object was composed of a cylinder of 80 mm in length and 80 mm in diameter, with uniform relative permittivity equal to 3.05 (contrast χ=2.05). It was the most difficult object of the database because of its large size and high contrast, and several methods failed to reconstruct this object [28,33,63,66], while many others produced poor quality results [27,29,61,62,64]. Figure 5 shows our reconstruction results. The shape and the contrast were adequately reconstructed: the contrast function was homogeneous inside the object and the estimated value at the center voxels was close to the true one, since the estimated permittivity is about 1.9. To the best of our knowledge, this reconstruction of Cylinder outperformed all other results previously reported in the literature.

On a broader perspective, Table 3 provides a heuristic assessment of the reconstruction quality for all objects in the Fresnel database. Each cell in the table contains a subjective comparison of our result with previously obtained ones for each object, based on the figures in the corresponding papers. We consider that the proposed method provided results at least similar to previous ones, and often better for the most difficult objects. The reader is referred to supplementary files associated with the paper, that depict graphical representations of all reconstructed objects and provide all reconstruction results as Matlab files.

Table 4 finally gives the reconstruction time for the five objects of the Fresnel database using either BiCGStab or block-BiCGStab. The time saved by block-BiCGStab becomes more important as the difficulty of the reconstruction problem increases (i.e., as objects become larger and/or more contrasted). For the most difficult Cylinder object, the reconstruction time drops from 60 h to 28.3 h, which means that block-BiCGStab is more than twice faster than BiCGStab.

## 6. Discussion

We proposed a reconstruction strategy based on an additive regularization framework, where optimization is performed via first-order optimization. Computations of the cost function and of its gradient were performed through a dedicated implementation of the block-BiCGStab algorithm for multiple RHS systems inversion. In all our experiments, such an implementation led to a significant reduction of the computation time (up to one half) compared to BiCGStab, the gain being all the more significant that the reconstruction problem is difficult and the computation time is high. The procedure proposed in this paper is therefore particularly attractive for problems dealing with high-frequency illuminations and/or highly contrasted objects, which are of great interest: high-frequency acquisitions contain more detailed spatial information about the object under study, yielding higher resolution, and highly contrasted objects may be encountered in many application areas, such as biomedical imaging. Consequently, the proposed procedure may contribute to reducing the computational burden of microwave imaging, which still represents a crucial issue in the development of practical applications.

Improved convergence properties of block-BiCGStab over its sequential counterpart also enable block-BiCGStab to reach a low residual level for each iterative system inversion. In our experiments, the tolerance threshold on the norm of the residuals was set to $10^{-6}$, whereas higher values are commonly used. More accurate solutions to the forward problem yield more efficient optimization steps; this may explain why the proposed procedure achieved satisfactory reconstruction results for the most difficult objects of the Fresnel database, for which no result of equivalent quality has been reported in the literature.

Exploiting the proposed block resolution strategy may also improve the efficiency of any other microwave imaging inversion method requiring the computation of similar quantities (i.e., the total fields from the incident fields), either based on additive [13] or multiplicative [33] regularization. This strategy can also be advantageously used whatever the choice of the optimization method, such as non-linear conjugate gradient [29,64], regularized Gauss–Newton [33], Levenberg–Marquardt [67], or distorted Born iterative method [27]. It would also be worth studying the numerical performance of the block-BiCGStab algorithm in different geometrical configurations concerning, e.g., data acquisition setups or discretization schemes. Finally, it may also be efficiently used in other applications involving the computation of the total electromagnetic field accounting for scattering, e.g., radar cross-section [48,68,69], eddy-current [26,70] or magnetotelluric [71] computations. Depending on the complexity of the problem, specific tuning of the block-BiCGStab algorithm could be required, for which the strategies proposed in this paper may help.

Further improvements may be brought by parallel implementation of the proposed procedure. In our experiments, multicore architecture was taken advantage of only by performing multithreaded operations at a low level within the block-BiCGStab loop. However, massive parallelization of the resolution of independent systems (with BiCGStab) reduces the computation time by exploiting parallelization at a higher level [33]. Indeed, the two parallelization levels could be combined into a block resolution procedure, where the multiple systems would be split into several subsystems to be solved in parallel. In [53], such an idea was successfully tested on an early version of block-BiCGStab. This remains a promising possibility to accelerate microwave imaging in the context of High Performance Computing, as well as parallelization capacities brought by GPU- or multi-GPU-based implementations.

## Figures and Tables

**Figure 1 sensors-20-06282-f001:**
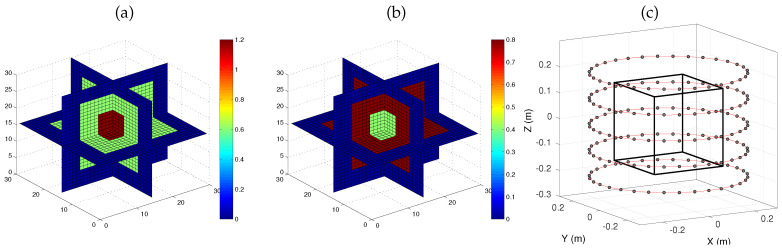
Real (**a**) and imaginary (**b**) parts of the contrast function of the simulated object (two nested cubes in air), with the corresponding discretization grids, and acquisition setup for the simulated object (**c**): the black cube represents the unknown volume *V* in which the object is placed. The gray circles represent the sources positions around the volume. The receivers are located at the same positions as the sources.

**Figure 2 sensors-20-06282-f002:**
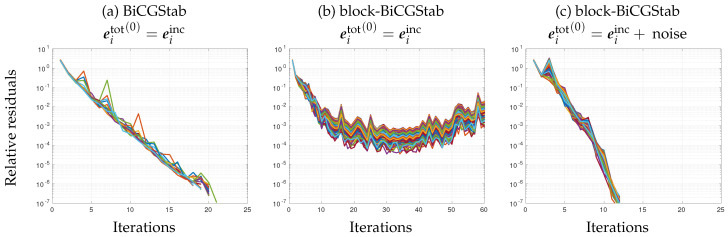
Resolution of 160 forward problems with biconjugate gradient stabilized (BiCGStab) (**a**) and with block-BiCGStab (**b**,**c**): evolution of the residuals for the 160 iterates. (**a**,**b**): initialization is set to the incident fields. (**c**) initialization of block-BiCGStab is perturbed with 50 dB circular Gaussian noise. Note the different abscissa scale on panel (**b**).

**Figure 3 sensors-20-06282-f003:**
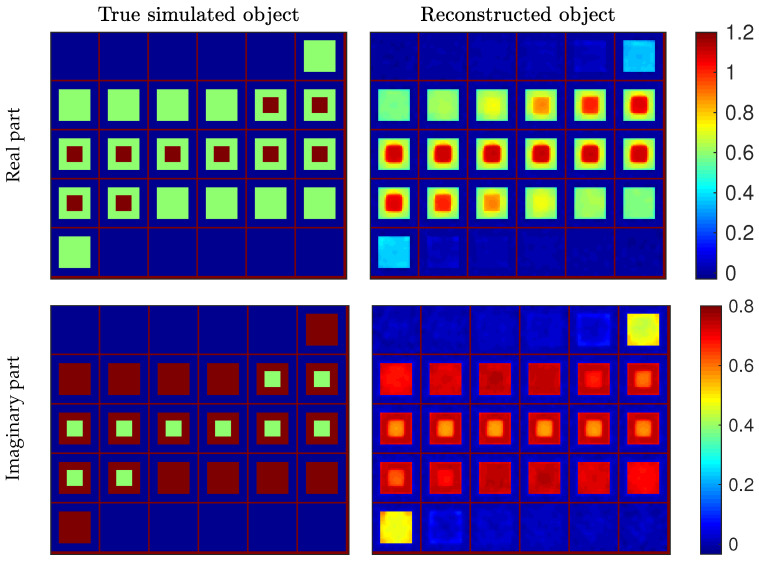
Reconstruction results (contrast values) for the simulated object defined in Section 4.1. Each panel represents the 30 slices (30×30 pixel images) of the three-dimensional object. True (**left**) and reconstructed (**right**) objects, with real (**top**) and imaginary (**bottom**) parts of the contrast function.

**Figure 4 sensors-20-06282-f004:**
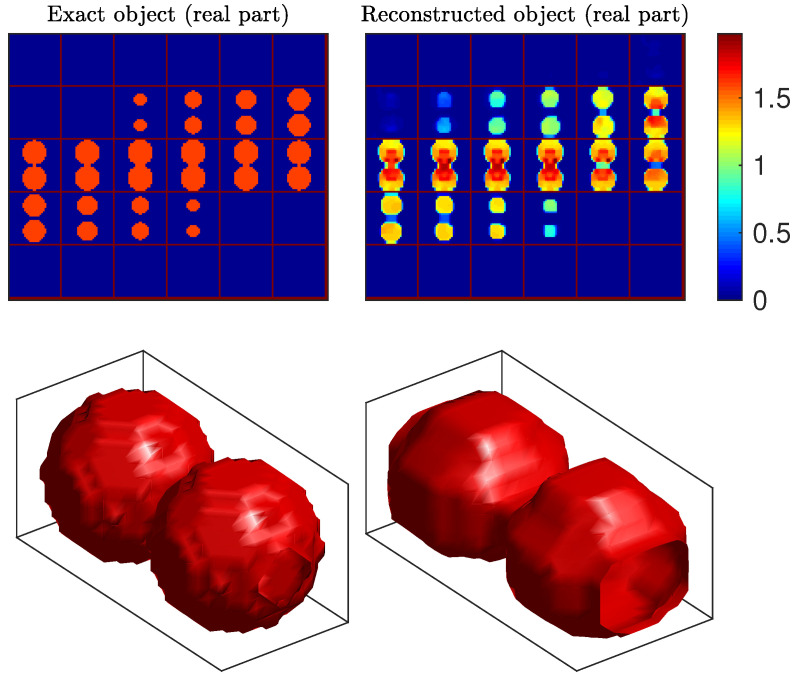
Reconstructed contrast function for the TwoSpheres object. Top: 30 horizontal slices of the true object (**left**) and the 30 slices of the reconstructed object (**right**). Bottom: isosurfaces of the true object discretized on the reconstruction grid (**left**), and of the reconstructed object (**right**), for contrast value 1.

**Figure 5 sensors-20-06282-f005:**
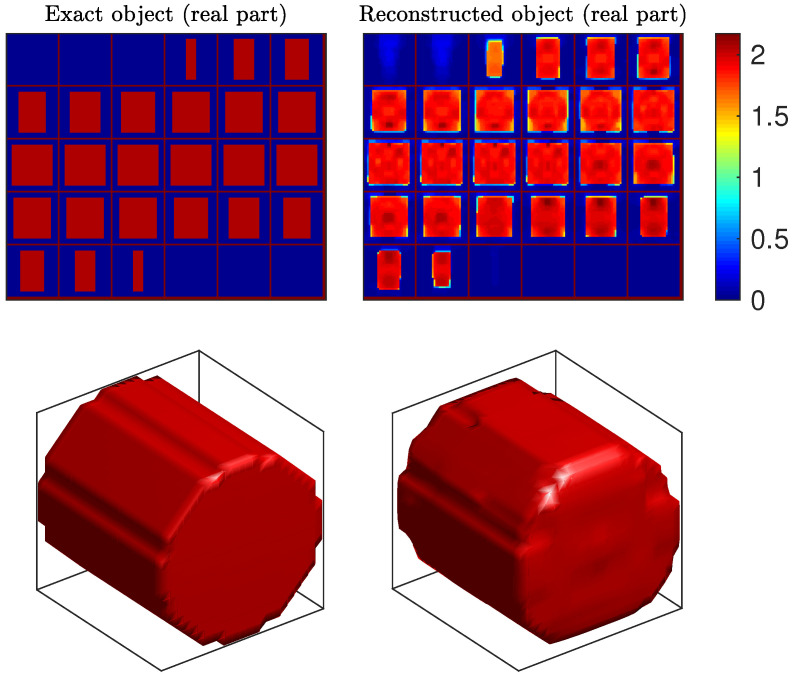
Reconstructed contrast function for the Cylinder object. Top: 30 horizontal slices of the true object (**left**) and the 30 slices of the reconstructed object (**right**). Bottom: isosurfaces of the true object discretized on the reconstruction grid (**left**), and of the reconstructed object (**right**), for contrast value 1.5.

**Table 1 sensors-20-06282-t001:** CPU time and number of iterations required for computing the 160 forward simulations with BiCGStab and block-BiCGStab for three different frequencies. Results are averaged over the 160 resolutions for BiCGStab and over 20 realizations of the random initialization for block-BiCGStab.

Frequency	CPU Time		Number of Iterations		CPU Time Ratio Block/BiCGStab
			[min, Average, max]		
	BiCGStab	Block	BiCGStab	Block	
1 GHz	116 s	110 s	[7 8.0 10]	[7 7.9 9]	0.95
2 GHz	187 s	138 s	[13 13.4 15]	[10 10 10]	0.74
3 GHz	245 s	164 s	[17 17.9 21]	[12 12 12]	0.67

**Table 2 sensors-20-06282-t002:** CPU time and number of iterations required for computing the 160 forward simulations with BiCGStab and block-BiCGStab for different contrast levels. Results are averaged over the 160 resolutions for BiCGStab and over 20 realizations of the random initialization for block-BiCGStab.

Contrast	CPU Time		Number of Iterations		CPU Time Ratio Block/BiCGStab
			[min, Average, max]		
	BiCGStab	Block	BiCGStab	Block	
0.25	105 s	86 s	[7 7.01 8]	[6 6 6]	0.82
0.5	152 s	120 s	[10 10.3 12]	[8 8.5 9]	0.79
1	245 s	164 s	[17 17.9 21]	[12 12 12]	0.67
1.5	361 s	217 s	[25 26.9 29]	[15 16.1 18]	0.60
2	482 s	271 s	[34 36.1 39]	[19 20.2 21]	0.56
2.5	610 s	331 s	[44 45.9 50]	[23 24.8 26]	0.54

**Table 3 sensors-20-06282-t003:** Subjective comparison of the obtained reconstruction results with previously published works on Fresnel database objects: “=”, “+” and “++” respectively mean that our reconstruction is of comparable, better, and much better quality. N/A means that reconstruction of the object is not presented in the corresponding paper.

Paper Ref.	TwoCubes	IsocaSphere	CubeSpheres	TwoSpheres	Cylinder
[63]	++	=	=	++	N/A
[29]	++	=	+	++	++
[28]	+	+	=	+	N/A
[64]	++	+	+	++	++
[61]	+	=	+	=	+
[27]	+	=	=	=	++
[33]	+	=	N/A	N/A	N/A
[66]	+	N/A	N/A	=	N/A
[62]	N/A	N/A	+	=	++

**Table 4 sensors-20-06282-t004:** Reconstruction time for the five objects of Fresnel database using BiCGStab and block-BiCGStab for linear system inversions.

Object from	Time	Time	Time Ratio
Database	BiCGStab	Block-BiCGStab	Block/BiCGStab
TwoCubes	2.33 h	2.07 h	0.89
IsocaSphere	4.38 h	3.58 h	0.82
CubeSpheres	3.21 h	2.60 h	0.81
TwoSpheres	8.60 h	6.41 h	0.75
Cylinder	60.0 h	28.3 h	0.47

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
