# Peer review of "Three-Dimensional Microwave Imaging: Fast and Accurate Computations with Block Resolution Algorithms"

_sensors, 2020, doi:10.3390/s20216282_

Round 1

Reviewer 1 Report

Multiple targets simulations should be carried out for this study. In addition, noise should be added for simulation tests to improve the accuracy of your algorithm.

Multiple contrasts for multiple targets should be tested for your method.

How you solve the forward equations for complex geometry?

For inverse, you use nonlinear or linear algorithm? Is there any normalization or regularization scheme used? 

Did you try use GPU to speed your calculation.

What is the lowest contrast and imaging resolution?

What about the quantitative information besides the structural information?

Can the recovered functional information be used for clinical diagnosis?

Reviewer 2 Report

  1. Figure 2 shows that the block-BiCGStab can effectively accelerate the convergence when 50-dB noise is added to the incident field. I am very confused about that, why the addition of such a high noise speeds up the convergence. Why add 50dB of noise, is it possible to add 20dB, 30dB or 10dB (how to choose the level of noise addition)? In the subsequent numerical tests, how was the initial value chosen?
  2. It can be seen from Table 3 that the proposed method has great advantages over the previously methods, but we don’t know what this comparison is? Is it any quantitative comparison of calculation errors or a comparison of intuitive results?
  3. Line 110, the authors said “Additive regularization is particularly efficient in the case of highly contrasted objects, whereas multiplicative regularization may not converge.” We can’t agree with it. The multiplicative regularization and additive regularization are two important TV regularization and they have their advantages. And multiplicative regularization avoids to select the regularization parameter empirically, which is also verified for the inversion of highly contrasted objects in [R1]. The statements should be revised accordlingly.

[R1] K. Xu, Y. Zhong and G. Wang, "A Hybrid Regularization Technique for Solving Highly Nonlinear Inverse Scattering Problems," in IEEE Transactions on Microwave Theory and Techniques, vol. 66, no. 1, pp. 11-21, Jan. 2018, doi: 10.1109/TMTT.2017.2731948.

  1. Please give the computational complexity analysis in the paper. Besides the comparison of computational cost,could give the comparison on the quantitave retrieval results between the block-BiCGStab and BiCGStab ?
  2. In chapter 4.4 and chapter 5, in the inversion, the frequency-hopping is used. How did you decide the iteration number for each frequency? Is there any strategy for it? For example,

Line 272, 20 iterations at each frequency are used for inversion.

Line 311, 10 iterations at each frequency are used for inversion.

Reviewer 3 Report

In the present manuscript, the Authors present a block-BiCGStab algorithm for solving three-dimensional microwave imaging problems. The paper is interesting and the topic is well suited for the Sensor journal. I have just some minor remarks that the Authors should consider.

-) In section 2.2, the Authors state that the vector e_scatt in eq. (6) has size N_r, i.e., equal to the number of receiving points. However, since it contains electric field samples, it should be composed by three components for every point, leading to a size 3N_r (as for the internal field in eq. (2). Is it a typo, or do the Authors consider only a single field component? In the latter case this should be specified in the formulation of the method.

-) How have the values of the parameters of the method used in section 4.4 and 5 (numbers of iterations in the intermediate frequency steps, regularization parameters, …) been chosen? Usually they could significantly affect the final results, at least some discussion on their selection should be added.

-) How has the “Subjective comparison” in Table 3 been performed? In order to be fair, it should be based on a quantitative analysis (e.g., of the reconstruction errors), otherwise I don’t think it is sufficient to “strongly support” the claim that “the proposed method provides results at least similar and often superior” to the others. The Author should either provide some quantitative information to support the comparison or reformulate such sentences.

Round 2

Reviewer 2 Report

The authors have good reponse to the comments.